# Conjugated Polymer-Based Nanocomposites for Pressure Sensors

**DOI:** 10.3390/molecules28041627

**Published:** 2023-02-08

**Authors:** Qin-Teng Lai, Qi-Jun Sun, Zhenhua Tang, Xin-Gui Tang, Xin-Hua Zhao

**Affiliations:** 1School of Physics and Optoelectric Engineering, Guangdong University of Technology, Guangzhou 511400, China; 2Department of Materials Science and Engineering, City University of Hong Kong, Hong Kong 518060, China; 3Department of Chemistry, South University of Science and Technology of China, Shenzhen 518060, China

**Keywords:** conjugated polymers, flexible pressure sensors, healthcare, electronic skin, artificial intelligence

## Abstract

Flexible sensors are the essential foundations of pressure sensing, microcomputer sensing systems, and wearable devices. The flexible tactile sensor can sense stimuli by converting external forces into electrical signals. The electrical signals are transmitted to a computer processing system for analysis, realizing real-time health monitoring and human motion detection. According to the working mechanism, tactile sensors are mainly divided into four types—piezoresistive, capacitive, piezoelectric, and triboelectric tactile sensors. Conventional silicon-based tactile sensors are often inadequate for flexible electronics due to their limited mechanical flexibility. In comparison, polymeric nanocomposites are flexible and stretchable, which makes them excellent candidates for flexible and wearable tactile sensors. Among the promising polymers, conjugated polymers (CPs), due to their unique chemical structures and electronic properties that contribute to their high electrical and mechanical conductivity, show great potential for flexible sensors and wearable devices. In this paper, we first introduce the parameters of pressure sensors. Then, we describe the operating principles of resistive, capacitive, piezoelectric, and triboelectric sensors, and review the pressure sensors based on conjugated polymer nanocomposites that were reported in recent years. After that, we introduce the performance characteristics of flexible sensors, regarding their applications in healthcare, human motion monitoring, electronic skin, wearable devices, and artificial intelligence. In addition, we summarize and compare the performances of conjugated polymer nanocomposite-based pressure sensors that were reported in recent years. Finally, we summarize the challenges and future directions of conjugated polymer nanocomposite-based sensors.

## 1. Introduction

Flexible sensors have attracted tremendous attention due to their potential applications in the fields of health monitoring, medical diagnosis, electronic skin (e-skin), and artificial intelligence [1,2,3,4,5,6,7,8,9,10,11,12,13,14,15,16,17,18,19,20]. In recent years, flexible sensors have made great progress in material selection, structure design, and practical application [21,22,23,24,25,26,27,28,29,30,31,32,33,34,35,36]. Currently, the most widely studied sensing materials include traditional silicon-based materials, flexible and stretchable polymers, and conductive carbon and metal nanostructures, including nanoparticles, nanowires, nanosheets, and nanofibers [37,38,39,40,41,42,43,44,45,46,47,48,49,50,51,52,53,54]. These developed sensing materials have paved the way for the improvements of sensor performances and practical applications. Although researchers have devoted great efforts to the development of flexible sensors, it is still challenging to mass produce wearable sensing devices with high sensitivity and a wide detection range for health monitoring, flexible display, and soft robots.

Polymers have excellent flexibility and boost the development of flexible sensors. In particular, CPs have excellent electrical properties because of alternating single and double covalent bonds for conducting electrons and their inherent delocalized π-electron adjustability ranges from semi-conductive to metal-conductive [55,56]. Apart from that, CPs have excellent mechanical flexibility and biocompatibility, which enables them to be excellent candidates for sensors [57]. Besides the above-mentioned properties, the CP-inherent π-conjugate systems are susceptible to hydrogen bonding, charge transfer, and dipolar-dipole interactions, allowing them to have greater functionality in pressure sensor applications, such as improving stability, sensitivity, and sensing ranges [58]. Because of the above-mentioned merits, CP-based sensors can detect small disturbances with high sensitivity. Therefore, it is of significance to investigate CP-based sensors, which can provide direction for the development of flexible electronics.

Among the flexible sensors, a pressure sensor is one of the most important components in flexible and wearable electronics; it can convert mechanical stimuli or deformation from the outside world into electrical signals. In this work, we mainly describe pressure sensors based on the following four transduction mechanisms—piezoresistive [59,60], capacitive [61,62], piezoelectric [63,64], and triboelectric (Figure 1) [65,66,67]. When a piezoresistive pressure sensor is subjected to an external force, the distribution and contact states of the internal sensing material change, resulting in a change in the corresponding electrical signal (Figure 1a). These sensors are simple in device construction, easy to fabricate, and involve easy signal processing. However, they have high power consumption and response hysteresis, leading to limitations in some applications [68]. As shown in Figure 1b, capacitive pressure sensors vary the capacitance of the device by changing the area of the dielectric layer or the distance between the electrodes on either side by external mechanical force. Piezoelectric pressure sensors are subjected to external forces of uneven charges within the material, which produces charge polarization (Figure 1c) [69,70]. Triboelectric pressure sensors can generate electrostatic charges due to the periodic mechanical energy of the positive and negative electrodes that contact and separate, depending on the frictional material of the different frictional electrode properties (Figure 1d) [71]. Piezoelectric and triboelectric pressure sensors can operate as self-powered pressure sensors without any external power supply. They can reliably detect dynamic pressure; however, the output signal is susceptible to interference from the external environment, and there are challenges in static pressure detection [72,73].

Thanks to the development of materials science and engineering, conjugated polymer nanocomposites have been extensively studied in the discipline of flexible electronics [74,75]. For example, π-conjugated conductive polymers (PEDOT, PANI, and PPy) have been successfully used as sensing layers or electrodes in flexible sensors [76,77,78]. They can form films independently or conductive polymers can be processed into nanostructured materials to be injected into other polymeric matrices (e.g., PDMS, PU, PVA) to form composite films with interpenetrating structures [79,80,81].

In this paper, we first briefly introduce the performance parameters and operating principles of piezoresistive, capacitive, piezoelectric, and triboelectric pressure sensors. Second, we review typical examples of pressure sensors based on conjugated polymer nanocomposites in terms of the device structure, preparation techniques, and performance. Then, we present the applications of pressure sensors based on conjugated polymer nanocomposites for healthcare, human motion monitoring, electronic skin, and artificial intelligence. In addition, we briefly summarize and compare the performance of conjugated polymer nanocomposite-based pressure sensors that have been reported in recent years. Finally, the remaining challenges for conjugated polymer nanocomposite-based sensors and the future development of high-performance sensor devices are summarized. We hope that the discussion in this paper will be helpful for the development of conjugated polymer nanocomposite-based sensors.

## 2. Parameters for Pressure Sensors

Pressure sensors are analytical devices that are relevant to daily life and scientific research processes; they involve acquiring and processing physical stimuli and converting them into corresponding signals according to certain laws. The pressure sensor device usually consists of a sensitive element and a conversion element. The sensitive element refers to the part of the sensor that directly feels what is being measured. Its response to the external stimulus is determined by multiple factors, such as the inherent characteristics of the sensing material, the structure of the device, and the morphology of the sensing layer. Therefore, scientists can improve the performance of sensors by synthesizing new composite materials, exploring new manufacturing processes of devices, constructing the microstructure of material morphology, and modifying the surface. It is important to make use of the inherent conductivity, and optical and electronic properties of conjugated polymer nanocomposites to prepare high-performance sensors.

Pressure sensitivity and the linear detection range are two key parameters used to evaluate the performances of pressure sensors. However, there is a certain contradiction between sensitivity and detection range. Sensitivity requires that the sensor undergoes significant changes in the internal material structure and distribution at small pressures, resulting in large changes in the corresponding electrical signal. The detection range requires the device to maintain circuit connectivity over a wide range of pressures. Currently, it is still challenging to prepare pressure sensors with both high sensitivity and a wide linear range. A major goal in the field of polymers is the creation of conjugated polymers and nanocomposites that combine high electrical conductivity and mechanical strength [82]. Consequently, conjugated polymer-based pressure sensors maintain an optimal balance between pressure sensitivity and detection range. In addition, the pressure response time, the limit of detection (LOD), durability, and mechanical stability are also parameters that need to be considered when developing pressure sensors.

### 2.1. Pressure Sensitivity

The pressure sensitivity (S) of a pressure sensor is expressed as the ratio of the corresponding output electrical signal value relative to the change in applied pressure. The sensitivity of a pressure sensor is expressed as S = (ΔA/A_0_)/ΔP, where A_0_ is the value of the initial electrical signal of the sensor (e.g., current, capacitance, voltage, etc.), ΔA is the relative change in the electrical signal, and ΔP is the change in applied pressure. Usually, the sensitivity is improved by reducing the initial electrical signal A_0_. For example, microstructures are introduced into the sensor to reduce the initial contact area [83,84]. The use of new composite materials and the design of microcracking and slip mechanisms contribute to the high sensitivity of the sensor even over a wide sensing range [85,86].

### 2.2. Linear Response Range

The linearity between relative output and external pressure over the entire pressure sensing range is very important. Typically, the entire pressure sensing range is generally described in terms of both low and high pressure. Most pressure sensors have a good response in the normal pressure range, but fail to respond in the very low-pressure region or the sensor output signal quickly saturates after a certain pressure. Therefore, the linear sensing range should be fully considered when developing the sensor, which can simplify the signal acquisition and processing process and broaden the application area of the device.

### 2.3. Response Time and LOD

The human skin has many receptor nerves that can respond quickly to small external stimuli and feel which parts of the body they come from. Response time and LOD are two other important parameters used to evaluate the performances of pressure sensors. Response time refers to the speed of the sensor’s response to real-time detection of external stimuli. The LOD reflects the detection accuracy of the sensor and indicates the minimum pressure that can be detected by the sensor. Applications such as monitoring subtle body movements and wrist pulses require a low limit of detection.

### 2.4. Stability

Stability means that the output characteristics of the pressure sensor remain consistent or do not exceed the allowable error range in a specific operating environment. In practical applications, it is a key factor for high-performance sensing. The stability of the sensor is determined by multiple factors. For example, the chemical and thermal stability of the sensing layer material, substrate selection, device structure, and packaging.

## 3. Advances in Pressure Sensors Based on Conjugated Polymer Nanocomposites

### 3.1. Piezoresistive Pressure Sensors

Piezoresistive pressure sensors are defined as converting a change in external pressure into a change in resistance. The sensor resistance usually consists of the interface contact resistance between the sensing material and the electrode, the contact resistance at the interface between multiple layers of sensing material, and the internal resistance of the sensing material and the electrode. The resistance R = ρL/S, where, ρ, L, and S denote the resistivity, length, and cross-sectional area of the sensor, respectively, has received a lot of attention from researchers because of its relatively simple structure, sensing mechanism, and easy signal acquisition. As shown in Figure 2, under the initial conditions, the sensor is in a high resistance state and the output current I_0_ is relatively small. At the same drive voltage, the output current (I_P_) increases by applying external pressure to the sensor. Thus, piezoresistive pressure sensors can measure external pressure through changes in resistance or output current. In past decades, conjugated conductive polymer sensing materials commonly used for piezoresistive sensors were: (poly(3,4-ethylenedioxythiophene):poly(styrenesulfonate) (PEDOT:PSS) [87], polypyrrole (PPy) [88], polyaniline (PANI) [89], etc. In this part, we will focus on the selection and structural design of conjugated conducting polymers to improve the performances of piezoresistive pressure sensors.

Wang et al. used silicon carbide sandpaper as a template to prepare PDMS with a hump-like microstructure, and then spin-coated PEDOT:PSS on its surface to form an active layer for piezoresistive pressure sensors [90]. Figure 3a shows the microstructure and sensing schematic for different heights and shapes. As the load pressure increases, the contact area between the microstructure on the sensor surface and the electrodes increases, resulting in a decrease in resistance. When the pressure sensor is under a higher load, the higher hump significantly increases the contact area between the hump and the electrode and also creates an additional contact path with the shorter hump, thus further reducing the resistance. Figure 3b shows the schematic diagram of the PEDOT:PSS/PDMS-based microstructure sensor preparation process and SEM images of four different sizes of the PDMS microstructure. Thanks to the high conductivity and stable chemistry of PEDOT:PSS and the PDMS hump-shaped microstructure, the sensor obtained a high sensitivity (of 851 kPa^−1^) and has been successfully applied in biomedical applications.

Conjugated polymer PEDOT:PSS has high electrical conductivity, π-bond interactions, and high molecular weight, which can be firmly adsorbed on the fiber surface as a binder and separate the mechanical and electrical properties of the fiber. Thus, wearable sensors based on PEDOT:PSS-coated fibers or textiles combine high sensitivity with excellent mechanical properties. As shown in Figure 4a,b, Wang et al. were inspired by spider villi to obtain PEDOT:PSS fibers with array microstructures by an ion-induced self-assembly technique [91]. The SEM images show that PEDOT:PSS-Cu^2+^ is uniformly distributed on the fiber surface, and the dense fluffy array completely encapsulates the PEDOT:PSS fiber. In addition, Figure 4f demonstrates that PEDOT:PSS fibers have good flexibility. Then, the PEDOT:PSS-Cu^2+^ fibers were integrated into the woven structure to prepare the tactile sensor (Figure 5). As shown in Figure 5a–c, the GF variation of the fiber during stretching and the cyclic stretch/release at a 15% strain demonstrate the high sensitivity and good stability of the sensor. Figure 5d–i show the change in resistance of the sensor at different bending angles, the change in resistance of the sensor under pressure testing, as well as the response time. The experimental analysis showed the excellent deformation response and long-term stability of the PEDOT:PSS-Cu^2+^ fiber-based tactile sensor.

The holder assembly of a flexible piezoresistive pressure sensor usually determines the strain range of the sensor. Conductive polymer composites with good mechanical properties and compressible three-dimensional (3D) structures are ideal materials for pressure sensors. Typically, 3D supports require excellent mechanical flexibility and high compressibility under applied compression forces, while conductive components require sensitive piezoresistive changes during compression/recovery. As shown in Figure 6a, Zhao and co-workers prepared a highly flexible conductive aerogel by mixing PEDOT:PSS with polyimide (PI) using freeze-drying and thermal annealing techniques [79]. Benefiting from the interaction of hydrogen bond and π–π stacking interaction between the highly conductive PEDOT:PSS and PI molecular chains, the aerogel has high electrical conductivity as well as excellent compressibility and linear piezoresistive response. In addition, Lu et al. prepared a sandwich-like poly(3,4-ethylenedioxythiophene) (PEDOT) composite nanosheet with high electrical conductivity, hydrophilicity, and redox activity on polydopamine-induced sulfonated graphene oxide (PSGO), as shown in Figure 6b [92]. Finally, PEDOT-PSGO nanosheets were doped into PAAM as conductive filler to develop a stretchable, conductive, and viscous hydrogel for the precise detection of physiological signals.

PANI, another conductive polymer, has been widely used as a sensing material for sensors because of its high electrical conductivity, easy synthesis, good chemical stability, and biocompatibility [80,93]. As shown in Figure 7a, Chang et al. obtained conductive and elastic backbones by in situ polymerizations of PANI on a porous melamine sponge, followed by deposition of monolayer MXene nanoflakes on the conductive sponge using a freeze-drying technique [81]. The MXene/PANI sponge-based sensor has a 3D interconnected conductive network and excellent mechanical properties. Due to the stable 3D conductive network of the PANI sponge and MXene sheet in close contact, the MXene/PANI sponge can maintain the complete conductive network under large area deformation. Therefore, the MXene/PANI sponge-based sensor has excellent sensitivity and a wide pressure linearity range. The sponge form of 3D graphene bulk material with stable physical properties and unique structural design has attracted extensive research interest in flexible flex sensors. Huang and co-workers synthesized a novel layered composite of conductive PANI nanowire arrays, and then prepared rGO-PANI sponges (rGPS) with ordered microstructure and good electrical conductivity by hydrothermal self-assembly freeze-drying method (Figure 7b) [89]. The rGPS exhibits excellent sensitivity and fast response in a wide detection range. Therefore, it can be applied to the detection of human physiological signals and small vibration forces.

Inspired by the roof stacked tile structure, the MXene/PANIF nanocomposite sensing layer with a tile-like stacked layer microstructure was prepared by coating MXene sheets and PANIF layer-by-layer on an elastic rubber substrate, as shown in Figure 8a [94]. PANIF and MXene are firmly connected due to rich hydrogen bonding and electrostatic interaction, and PANIF as a bridge connecting MXene greatly improves the effective conductivity. In addition, the MXene and PANIF layers overlap each other to provide more conductive pathways to ensure the continuity of the conductive path. Benefiting from the tile-like stacking structure of MXene/PANIF, the strain sensor has a crack extension sensing mechanism and a sliding sensing mechanism during the stretching process, which enables the sensor to have high sensitivity over a wide strain range. Textiles can replace polymers (e.g., TPU, PDMS, and Ecoflex) as substrates for sensors because of their excellent flexibility, breathability, and ability to be prepared in large areas [95,96,97]. Textile-based strain sensors are required for various applications with high sensitivity, wide strain range, and long-term stability under severe wear and tear or multiple wash cycles. A textile sensor based on reduced graphene oxide was prepared by Xu et al. [98] This strain sensor has high tensile properties, wash resistance, and durability but low sensitivity. Therefore, Prof. Tan and her research group reported a PANI/rGO/PDA/LC-based strain sensor [77]. As shown in Figure 8b, PANI and reduced graphene oxide (rGO) were deposited on polydopamine PDA-modified lycra cotton (LC) substrates using in situ polymerization and impregnation reduction methods, respectively. The strain sensor has excellent sensitivity and high stretchability due to the synergistic effect of PANI and rGO in the LC substrate stretching and release process. In addition, the sensor shows a good response in human motion detection, providing a new research strategy for textile-based wearable sensors.

PPy, a highly conductive, easily synthesized, and extremely stable conjugated polymer has proven to be an effective way to extend its stretchable applications by blending it with flexible polymers to prepare functional polymer composites [99,100,101,102]. For example, Yang et al. used oxidative polymerization to grow PPy membranes with wrinkled microstructures on PDMS substrates (Figure 9a) [100]. Figure 9b shows the three-scale nested wrinkled microstructure of the PPy film, where the third surface wrinkle is formed by further heating/cooling treatment to increase the wavelength of the fold. Thanks to the microstructure of the PPy membrane surface, the sensor has a high sensitivity in the low-pressure range, but its sensitivity drops to 0.5 kPa^−1^ at pressures above 1 kPa and the entire sensing range does not exceed 2 kPa. The low sensitivity in the high-pressure range and the small sensing range limit its application in certain areas. Therefore, Li and colleagues used PPy sponges with multiscale porous structures to extend the sensing range and improve pressure sensitivity [101]. The preparation process and morphological evolution of porous PPy composites are shown in Figure 9c,d. The main sensing mechanism of the sensor is the change in the contact state of the porous PPy composite material caused by the force deformation in the hollow structure. With this large size structure, the hollow skeleton of the sponge compresses rapidly in the low-stress range resulting in a dramatic change in resistance. As the pressure is gradually increased, the contact area between the sponge skeletons increases, creating additional conductive paths, which can result in maintaining conductive paths over a wide range of stresses. As a result, the sensor remains highly sensitive over a wide sensing range.

### 3.2. Capacitive Pressure Sensors

For a capacitive pressure sensor, the capacitance is defined as C = εA/d, where ε, A, and d are the dielectric constant, the contact area between the electrodes, and the distance between the two electrodes, respectively. As shown in Figure 10, the capacitive pressure sensor works by changing the contact area of the dielectric layer and the distance between the electrodes on both sides by external pressure to cause a change in capacitance. In the initial state, the sensor has a small initial capacitance (C_0_) due to the distance between the two electrodes. When an external force is applied, the dielectric layer is compressed, the internal components are in closer contact, and the distance between the two electrodes is reduced, increasing the capacitance to C_p_. Therefore, the capacitive pressure sensor can reflect the external pressure change by the change of capacitance ΔC, and its sensitivity can be defined as S = (ΔC/C_0_)/(ΔP), and ΔP is expressed as the pressure change.

In recent years, capacitive pressure sensors based on conjugated polymers have been widely reported [103,104]. For example, a stretchable interleaved capacitive strain sensor based on carbon nanofiber/polyaniline/silicone rubber nanocomposite was developed by Hajghassem et al. [105]. In their work, highly conductive PANI was first coated with carboxyl-functionalized carbon nanofibers (CNFs) to prepare functionalized CNF-PANI composites, mixed with silicone rubber, and finally transferred to a mold to prepare electrodes with nanocomposite structures. This interfinger-type capacitive strain sensor has high sensitivity and good linear strain range.

On the other hand, a touch capacitive sensor is constructed by forming a plate electrode with air as the dielectric medium and woven fibers. As shown in Figure 11a, the yarn is coated with a conductive conjugated polymer PEDOT:PSS and a perfluoropolymer dielectric film cross braided onto the nylon fiber, which in turn causes a change in capacitance through the external pressure applied at the point of the crossed fiber [106]. Thanks to the unique structure of the fiber, a pressure-sensitive capacitive sensor array was developed with a high spatial resolution for precise detection of pressure distribution. Takamatsu et al. coated PEDOT:PSS on textiles to obtain electrode patterns with PDMS as an insulating layer and template to construct a textile-based wearable keyboard (Figure 11b) [107]. Then, a human finger touching the textile forms a pair of electrodes with the PEDOT:PSS/PDMS structure for capacitive sensing. The change in capacitance is detected by the amount of pressure of the finger pressing the keyboard.

The above textile-based capacitive pressure sensors all produce capacitance changes by using the human finger as a counter electrode and reducing the distance between the electrodes. In addition, by changing the contact area between the dielectric layer and the electrode, the geometry of the dielectric layer and the magnitude of the dielectric constant lead to a change in capacitance. Li and colleagues developed a flexible capacitive sensor with a through-hole for high sensitivity and a wide detection range [108]. As shown in Figure 11c, a PDMS film with air gap and high porosity (agp-PDMS) is sandwiched between two flexible PPy/filter paper composite films, where the PPy/filter paper composite film serves as the electrode of the sensor and the agp-PDMS film is the dielectric layer. The SEM image on the left side of the figure shows that the conjugated polymer PPy forms a continuous conductive film on the filter paper fibers and retains the original porous–rough structure of the filter paper. The SEM image on the right side clearly shows the different pore sizes of the porous PDMS layers. From the plot of pressure versus compressive strain at the bottom of Figure 11c, it is clear that the PDMS film with high porosity has high compressibility and its compressive strain at 1000 kPa pressure is several times higher than that of PDMS without porous structure. The PPy/filter paper composite film electrode surface rough structure and high porosity PDMS dielectric layer result in excellent sensing performance of capacitive sensors and are successfully applied to human motion monitoring and spatial mapping of pressure distribution. In conjunction with what is described in this part, capacitive pressure sensors have at least three sensing modes: (1) varying the distance between electrodes, (2) varying the contact area between the dielectric layer and the electrodes, (3) varying the geometry of the dielectric layer and the magnitude of the dielectric constant. Therefore, these factors can be used in the development of capacitive sensors to change the material and the structure of the device and, thus, improve its performance of the device.

### 3.3. Piezoelectric and Triboelectric Tactile Sensors Based on Conjugated Polymer Nanocomposites

Since Prof. Wang reported piezoelectric nanogenerators (PENG) based on ZnO nanowire arrays in 2006, research on piezoelectric and triboelectric systems have inspired academic research in the field of future self-powered haptic sensor systems and wearable electronics [109,110]. These sensors do not require an external voltage supply and can perform highly efficient low-frequency mechanically triggered energy conversion, thereby detecting external pressure changes through changes in voltage. Piezoelectric pressure sensors are designed to produce a piezoelectric effect by applying external mechanical pressure to some piezoelectric material, which generates a spatially positive and negative separated charge. Electric charges accumulate at the ends of the material to form electric dipoles. The relative displacement of the anions and cations leads to the formation of a piezoelectric potential. Triboelectric pressure sensors can effectively convert mechanical energy, especially low-frequency mechanical energy, into electrical energy by generating frictional electrical effects through frictional contact between different materials. The operating principles of piezoelectric and triboelectric pressure sensors are shown in Figure 12. These sensors have been widely used in dynamic pressure detection due to their self-powered feature [25,111,112,113]. However, they usually experience attenuation of the voltage signal when measuring static pressure. Therefore, it is still challenging to develop self-powered pressure sensors that can detect both static and dynamic pressures.

Due to the excellent electrical properties of conjugated polymer nanocomposites, PEDOT:PSS inherent π-conjugated system, susceptible to hydrogen bonding, charge transfer, and dipole-dipole interactions, has been widely used in piezoelectric and triboelectric pressure sensors. For example, Yang and his research group have developed a self-powered and multifunctional sock that uses a hybrid mechanism of piezoelectricity and triboelectric [114]. As shown in Figure 13a, the sock is integrated with a PEDOT:PSS-coated fabric triboelectric nanogenerator (TENG) and lead zirconate titanate (PZT) piezoelectric chip that enables energy harvesting and acts as a tactile sensor to sense various physiological signals and contact force analysis. The bottom of Figure 13a shows the PEDOT:PSS-coated fabric with polytetrafluoroethylene (PTFE) film collecting energy through contact separation mode, while the PZT piezoelectric chip acts as a force sensor to analyze the gait pattern and physiological signals of human motion. They report multifunctional smart socks with energy harvesting, motion monitoring, and healthcare, which provide an effective way to develop wearable devices with multifunctional smart systems. PEDOT is widely adopted as a coating or printable electrode with high conductivity due to its ease of synthesis, high conductivity, sufficient flexibility, and biocompatibility [115,116,117,118]. An all-organic piezoelectric nanogenerator (OPNG) with a multilayer structure was reported by Maity et al. [119]. Figure 13b shows the flow chart of the manufacturing process of the piezoelectric generator. Multilayer polyvinylidene fluoride (PVDF) nanofiber (NF) mats were first prepared by electrostatic spinning technique, and then organic conductive polymer PEDOT electrodes were deposited on PVDF NFs using gas-phase polymerization. The conductive ribbon is then attached to both electrodes and laminated with polypropylene (PP) film. Finally, the PEDOT-coated PVDF NFs felts were encapsulated with PDMS to prepare OPNG. The piezoelectric electrostatic spun PVDF NF mats with multilayer structure exhibit high voltage electrical properties due to the fiber interactions inside the mats and the synergistic effect of the PEDOT coating. OPNG can efficiently convert the mechanical energy of human finger movements into electrical energy and drive LEDs. OPNG exhibits excellent ultra-sensitivity to human motion and household machine vibration and is important as a sensor for measuring pressure. Hu and co-workers report a conductive piezoelectric nano-resistive network that can replace conventional sensor arrays for pressure sensing and localization [120]. As shown in Figure 13c, piezoelectric nano-resistance networks were established by preparing polyaniline/acrylonitrile (PANI/PAN) nanofibers (NFs) constructed by an integrated electrostatic spinning technique. Due to the 3D conductive network formed by the interaction of PANI molecular chains and PAN molecular chains in PANI/PAN NFs, when pressure is applied to PANI/PAN NFs, the internal conductive fibers will deform and become polarized, thus generating electrical signals. When the pressure is released, the induced charge flows in reverse during the return of the PANI/PAN NF to its initial state, resulting in a reverse signal. In their work, the PANI/PAN nano-resistance network is designed as a structure of Wheatstone bridge, and by detecting the output voltage distribution in each region, the PANI/PAN electrostatically spun nanofiber membrane can easily sense the pressure position. This piezoelectric nanoresistor network has great promise for tactile sensing, smart fabrics, wearable electronics, and other applications.

Conjugated conducting polymer nanocomposites (e.g., PEDOT, PPy) are also widely used as response layers, interconnects, and electrodes for frictional electrical pressure sensors [121,122,123,124,125]. Among them, PEDOT is easy to synthesize and exhibits good stability. Its aqueous dispersion PEDOT:PSS has relatively high ionic conductivity, specific capacitance, and biocompatibility, allowing the manufacture of smart textiles for energy harvesting and sensing using PEDOT:PSS coating [126,127]. A highly variable multi-arch frictional electric strain sensor was prepared by fixing a stretchable arch PEDOT:PSS-coated fabric on a silicone rubber substrate as shown in Figure 14a [128]. Due to this degree variable multi-arch structure, the sensor has a wider strain sensing range and has been successfully used for human activity monitoring, robotic control, and gas concentration sensing self-powered sensing applications. Zhen et al. prepared a transparent and stretchable TENG based on PEDOT:PSS electrode (WP-TENG) [129]. As shown in Figure 14b, a PDMS elastomer with a transparent and stretchable surface is prepared as a substrate. Next, PEDOT:PSS film is scraped on the PDMS surface. The mask micro-mask version is then used to determine the thickness of the PEDOT:PSS film and construct wrinkled microstructures by adjusting its layer number in a controlled manner. Finally, electrodes and encapsulated devices are inserted to obtain the WP-TENG. Benefiting from the special structural design of WP-TENG, its strain sensing range can reach 100%, and its good output performance in single electrode mode can easily drive electronic devices. In addition to energy harvesting, the WP-TENG acts as an active sensor that can be attached to the human skin to monitor human motion and spatial pressure distribution mapping.

PPy has high conductivity and pseudocapacitance, easy synthesis, and good adhesion, and is widely used in energy storage, and wearable sensors [130,131,132]. Prof. Wang and his research group reported a 3D network-like PPy/PDMS-based TENG [133]. Figure 14c shows the process flow diagram for the preparation of PPy/PDMS foam using the template method. By combining porous PPy with PDMS, the output performance of TENG is significantly improved and exhibits excellent flexibility. In addition, based on the coupled frictional electric effect and electrostatic induction, the TENG can combine static and dynamic pressure and acceleration detection, providing a new research idea for the development of self-powered tactile sensors.

## 4. Recently Reported Performance of Conjugated Polymer Nanocomposite-Based Pressure Sensors

In pressure sensors, the main conjugated polymer nanocomposites used are PEDOT: PSS, PANI, and PPy. Table 1 lists the recently reported pressure sensors based on conjugated polymer nanocomposites. The sensitivity, response range, and response time of pressure sensors are important for the selection of conjugated polymer nanocomposite materials for pressure sensor devices. Therefore, the relevant parameters of the pressure sensors are shown in Table 1 for comparison. Furthermore, to show the advance of conjugated polymer-based pressure sensors, we compare their device performances with those of the state-of-art pressure sensors based on other polymer composites as shown in Table 2.

## 5. Application of Conjugated Polymer Nanocomposite Based Pressure Sensors

### 5.1. Healthcare and Human Motion Detection

A growing number of studies have shown that pulse is an important indicator of cardiovascular disease and that a pulse that is too fast or short greatly increases the risk of coronary cardiovascular disease. Therefore, patients with cardiovascular disease must pay attention to their heart rate and manage it well. Pulse diagnosis is one of the most characteristic diagnostic methods in Chinese medicine. The pathological changes of human arterial blood vessels can be obtained through the pulse beat of the wrist pulse. The high-sensitivity pressure sensors are capable of accurately detecting vibrations generated by wrist pulses and can convert the pressure signal into an electrical waveform map of the radial artery via a conversion device. The analysis of pulse waveforms provides useful information for the diagnosis and treatment of cardiovascular diseases. For example, Wang et al. developed a PEDOT:PSS/PDMS-based high-sensitivity pressure sensor [90]. As shown in Figure 15a, the sensor is fixed on the wrist to detect the human pulse beat, collecting pressure information and converting it into an output current signal (Figure 15a_i_). The amplified pulse waveform in Figure 15a_i_ is shown in Figure 15c. It can be seen from the waveform graph that it has compression peaks and diastolic pressure lines typical of pulse maps. Figure 15a_ii_ shows the pulse waveform of a pregnant woman. The sensor can clearly distinguish the difference in pulse signal between pregnant and normal people with increased intravascular volume. In addition, as shown in Figure 15b, the pulse pressure signal was detected by active sensing with the sensor embedded in the fingertip glove, and the pulse waveforms were successfully acquired for normal (Figure 15b_i_) and pregnant women (Figure 15b_ii_).

Due to the excellent flexibility of pressure sensors based on conjugated polymer nanocomposites, they can be attached to human skin to monitor human motion. Zhan and co-workers reported a graphene (rGO)/PANI/thermoplastic polyurethane (TPU) mat (GPTM) assembly as a strain sensor [157]. In their work, PANI nanomaterials are coated on TPU mats as conductive conjugated polymers and synergize with rGO with high conductivity and high specific surface areas to enhance sensor performance. The sensor exhibits high sensitivity and a wide strain range to detect bending forces and accurately distinguish complex human movements. As shown in Figure 15d, the sensors are fixed at different locations on the human body to detect human movements, including small movements (e.g., articulatory and pulse vibrations, finger, wrist, and arm flexion deformations), as well as large movements (e.g., leg lifts and squats). Prof. Park and his research team have proposed a PANI/MXene/Polyacrylonitrile (PAN)-based hybrid nanofiber thin film sensor [158]. Benefiting from the rough structure of PANI-nanospin and fiber film surface, the sensor has a wide response range and high sensitivity. Figure 15e shows a schematic of the structure of the sensor and the wearable application. The sensors are fixed on the human wrist and neck to monitor the physiological signals of human movement in real time through a wireless system on a mobile platform. These studies show that conjugated polymer-based sensors have great potential for applications in wearable electronics and health monitoring.

### 5.2. E-Skin Application

Electronic skin is a flexible electronic device that mimics human skin. It can be as thin, soft, and stretchable as skin, and can easily fit on the surface of human skin or robots to feel external environmental stimuli, such as pressure, temperature, and object shape to achieve haptic perception [159]. Among the many sensing functions, pressure and temperature sensing are particularly important. Pressure sensing helps to monitor and control the applied pressure signal and to sense the shape of an object through contact. On the other hand, temperature sensing helps devices identify the temperature of objects and to detect physiological signals of living organisms. The selection of sensing materials plays a key role in the preparation of flexible electronic skins. The sensing material is required to be able to guarantee accurate pressure detection despite high stretching and even multimodal sensing capability. Conductive polymer PEDOT:PSS has good biocompatibility and remarkable thermoelectric properties; it can be used as an electronic skin tactile sensor preparation. Zhao et al. prepared a stretchable electronic skin based on PEDOT:PSS/single-walled carbon nanotube (SWCNT) electrodes [139]. As shown in Figure 16a, the sensor consists of a top and bottom PEDOT:PSS/SWCNT hybrid electrode array and a PDMS dielectric layer in the middle. Figure 16b shows the transmittance curves and surface morphology of the sensor, and the PEDOT:PSS/SWCNT electrodes embedded in the elastic PDMS greatly increase the transparency and stretchability of the device. The prepared electronic skin adheres to the hand intact under various deformations such as folding, twisting, and stretching and has excellent mechanical properties (Figure 16c,d). In addition, the e-skin has multi-point sensing of spatial pressure distribution and shape recognition in both stretched and non-stretched states. As shown in Figure 17a, different objects such as coins, beans, and paper stars are placed on the surface of the electronic skin, and their output capacitive signals can identify the weight of the objects. Figure 17b shows the spatial pressure distribution of the Z-object for the e-skin in the original state and at 20% tensile strain. It is shown that the e-skin exhibits excellent spatial recognition ability both in the initial state and in the stretched state. In addition to the spatial recognition capability, the e-skin also can recognize the shape of an object without touching it. As shown in Figure 17c, the device can capture the 3-dimensional information of the approaching finger when the finger is close to the e-skin. Not only that, but the e-skin can also clearly monitor the trajectory of the approaching finger (Figure 17d). Electronic skin has shown considerable application in the fields of flexible robotics, human prosthetics, and artificial intelligence.

In practical applications, the ability of electronic skins to recognize the temperature of objects is particularly important. Environmental sensors and haptic e-skins are applied by monitoring external temperature changes and collecting physiological signals from living beings. Gao and co-workers designed a multifunctional dual-mode pressure sensor array based on PEDOT:PSS/CNT@PDA@PDMS foam to be fabricated [142]. Since this e-skin has independent temperature and pressure sensing capabilities, it can generate stable voltage using the temperature difference between human skin and the surrounding environment as a self-powered wearable e-skin system. As shown in Figure 17e–g, a cup of hot water and a cup of cold water, as well as a finger touching the surface of the electronic skin, can accurately identify the external temperature change and pressure distribution. These results indicate that the sensor array has great promise for applications in electronic skin wearable devices and biosignal detection.

### 5.3. Smart Wearable Fabrics and Human–Machine Interaction

Textile-based wearable electronics with light mass, excellent flexibility, and mechanical properties have important research implications in the field of future wearable human–machine interfaces. Gestures are considered a linguistic communication technique that is intuitive, easy to use, and can be used as a special language to communicate with computers or robots. For example, a haptic sensor based on PEDOT/polyester (PS) polymer conductive fibers was reported by Eom et al. [160]. As shown in Figure 18a, the PEDOT/PS sensor is implanted in each finger of the glove and connected to a communication control system to read the output voltage of the sensor. Figure 18b illustrates the recognition of hand gestures by a wearable human–machine interface device. The device can perform predefined gestures and then communicate the corresponding characters through a wireless communication system, which finally leads to a corresponding change in the output voltage. Similarly, a smart textile self-powered sensor based on PEDOT:PSS coating was prepared by He et al. [128]. As shown in Figure 18c, the sensor is attached to the finger to monitor the sign language movements made by the hand, and the relatively significant change in the output voltage signal for each movement indicates that the sensor successfully captured the corresponding sign language movement. In addition, the sensor precisely controls the robot hand to make the corresponding movements through human gestures and grasping motions (Figure 18d).

Wearable textiles may experience chemical attacks from the external environment (e.g., corrosion from sweat, and oil stains), wear and tear, and severe wash cycles. Therefore, textile-based sensors must have mechanical robustness under severe wear and tear or multiple wash cycles. Xiao and his research team prepared smart wearable e-textiles by designing a “steel-concrete” layered structure of nanocomposite fibers [144]. As shown in Figure 19a, the conductive carbon nanotube “steel” network is first coated on the nanofiber surface and then a ppy-pda-perfluorodecyltrifluorotrioxysilane (PFDS) polymer layer is introduced to immobilize the carbon nanotubes, resulting in a superhydrophobic textile flexible sensor. Thanks to a polymer coating on the textile surface and a special structure, the sensor can accurately detect a range of human movements and physiological signals underwater. In their work, the hydrophobicity and corrosion resistance of the sensor was verified using a range of different chemical compositions. In addition, under the protection of polymer “concrete”, the textile-based sensor was machine washed for three cycles and subjected to several hours of agitated washing with negligible changes in sensitivity. As shown in Figure 19b, they developed a smart glove by embedding the sensor at the finger joint of the glove. The smart glove combined with the information acquisition and processing system can synchronize the human hand movements to control the corresponding movements of the robot hand. The smart gloves can control the manipulator to make the “I love you” gesture in the water (bottom of Figure 19b). This shows that textile-based wearable sensors have great commercial applications in the field of wireless communication and wearable human–machine interaction.

### 5.4. Sensor Applications in Robot-Training Systems and Robotic Tactile Sensing

The human brain can process complex tactile information and manipulate objects rationally because of the distribution of neurons, receptors, and synapses in the human sensory system [161]. These human capabilities provide a learning training set for the robot (Figure 20a) [144]. As shown in Figure 20b, the “steel-concrete” layered structure-based sensor finger sleeve developed by Xiao et al. can successfully collect empirical data on human grip and recognize the response to grasp different objects. Figure 20c shows the robot learns to grasp eggs by reading human-sensing information.

In human daily life and industrial manufacturing, sensors need to mimic biological skin to perceive the surface texture of objects. Therefore, robotic haptic sensing is of great research importance. Zhan and co-workers used an electro-gel method to prepare a 3D MXene/PEDOT:PSS composite aerogel (MPCA)-based haptic sensor [140]. In their work, the high electrical conductivity and flexibility of the aerogel are ensured by introducing the conjugated conducting polymer PEDOT:PSS. As shown in Figure 21a, they developed an MPCA-based high-resolution haptic sensor microarray and fixed it as an artificial tactile interface on the robot’s fingertips. The artificial tactile interface can directly perceive tactile stimuli from human fingers and can precisely distinguish spatial pressure distribution (Figure 21b). Thanks to the high sensitivity of the sensor, the artificial tactile interface perceives the texture roughness of the object surface by sliding over it. As shown in Figure 21c, the manual tactile interface was used for Braille recognition. From the corresponding pressure distribution diagram, it can be seen that the entire column of sensors can be accurate for the letter text corresponding to each Braille bump. Research shows that MPCA-based high-resolution tactile sensors have a wide range of applications in robotic tactile sensing and human–robot interaction.

## 6. Conclusions

In this paper, we review the recent progress of flexible pressure sensors based on conjugated polymer nanocomposites. Conjugated conductive polymers (PEDOT:PSS, PANI, PPy), also known as intrinsically conductive polymers, are widely used in a variety of pressure sensors, including capacitive, piezoresistive, piezoelectric, and triboelectric pressure sensors, due to their inherent π-conjugation system, electronic properties, and ease of synthesis. This paper presents and discusses the sensing mechanisms of pressure sensors based on different operating principles. Secondly, typical examples of different types of pressure sensors based on conjugated polymers (PEDOT:PSS, PANI, PPy) are reviewed. Finally, we discuss the applications of conjugated polymer-based pressure sensors in healthcare, human motion, electronic skin, smart wearables, haptic sensing, and artificial intelligence. Conjugated polymer-based flexible pressure sensors were realized with both high conductivity and mechanical properties, and their wide range of applications is demonstrated in the work reported above.

As presented in this paper, conjugated polymers PEDOT:PSS, PANI, and PPy have been widely used as sensing layers for flexible pressure sensors and have proven successful in a variety of applications. Benefiting from the electronic properties, molecular structure, ease of synthesis, and biocompatibility of CPs, CP-based sensors can sense very small perturbations and have high sensitivity. Therefore, many flexible sensors based on various CPs have made good research progress in terms of sensing performance and applications through the rational design of materials and device structures, providing new research ideas for preparing flexible pressure sensors and promoting the development of flexible sensors in commercial applications. In the past decades, great progress has been made in the controlled nano-assembly of conjugated compounds. However, our research on conjugated polymer materials is still in its infancy. At present, some conjugated polymer-based structural molecular and electrical properties have not been studied deeply enough, and relatively few types of materials (PEDOT:PSS, PANI, PPy) have been developed or matured for applications. To meet the diverse needs of practical applications, more types of conjugated polymers need to be developed for pressure sensors and to further improve the overall performances of such sensors, such as sensitivity, linear response range, response time, minimum detection limit, and stability. To achieve this goal, we offer the following ideas to provide some thoughts for continuing research in this area.

First, with the development of materials science and engineering technology, new structures of conjugated polymer nanocomposites are being developed and prepared by using new technologies and methods. Therefore, this will provide scientists with more new materials and structures to design sensors with diverse uses and better performance. On the other hand, these new material development techniques provide new avenues for sensor preparation. For example, the preparation of pressure sensors with both high sensitivity and a wide linear range, and pressure sensing devices with multimodal responses are urgent issues that need to be addressed.

Second, although many engineering techniques are used to prepare conjugated polymer sensing materials, most of the current work still has some shortcomings. There is still much work to be done to ensure the sensitivity, responsiveness, and durability of conjugated polymer-based pressure sensors. In addition, mass production of compatible and cost-effective conjugated polymer-based pressure sensors remains challenging. For example, we can design porous sensing layers or build microstructures on the surface, and these strategies can effectively change the contact resistance between the sensing materials and the electrodes, thus improving the sensitivity of the sensors. Therefore, in the future, we will have to develop new devices by developing new forms and perfecting the preparation technology of sensing materials, to finally realize the improvement of the performance and intelligent application of conjugated polymer-based sensors.

Third, integrating pressure sensor devices with other sensory sensor devices and power supplies in a single chip is the ultimate goal for flexible wearable sensing devices. This is important for applications in wearable pulse sensing devices, electronic skin, tactile sensing, artificial intelligence, etc. However, achieving this goal faces many problems, i.e., data acquisition, analysis, and conversion into electrical signals or multidimensional data are complex and difficult tasks. In terms of preparation technology, there are greater difficulties in integrating a series of different sensing devices and power supplies onto a chip with a limited area. Therefore, developing flexible conjugated polymer materials with excellent stretchability and availability and integrating hardware and software will help build sensing systems with excellent performance, ease of use, and versatility. We strongly believe that pressure sensors based on conjugated polymer nanocomposites will be commercially available shortly. Meanwhile, we hope this review can provide new research ideas and perspectives to researchers.

## Figures and Tables

**Figure 1 molecules-28-01627-f001:**
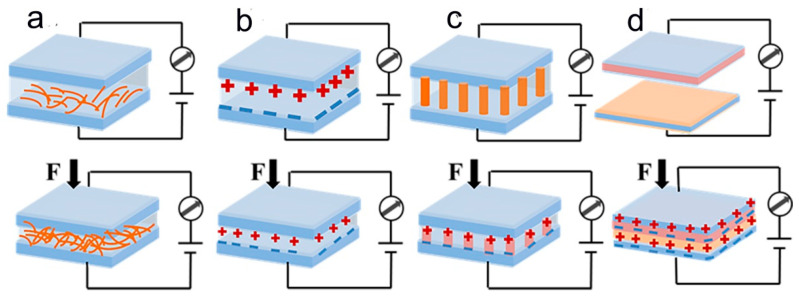
Schematic diagrams of four typical transduction mechanisms: (**a**) piezoresistive, (**b**) capacitive, (**c**) piezoelectric, (**d**) triboelectric.

**Figure 2 molecules-28-01627-f002:**
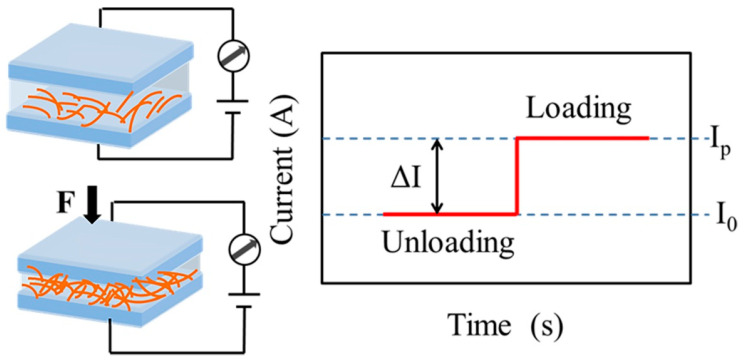
Work principle of piezoresistive pressure sensors.

**Figure 3 molecules-28-01627-f003:**
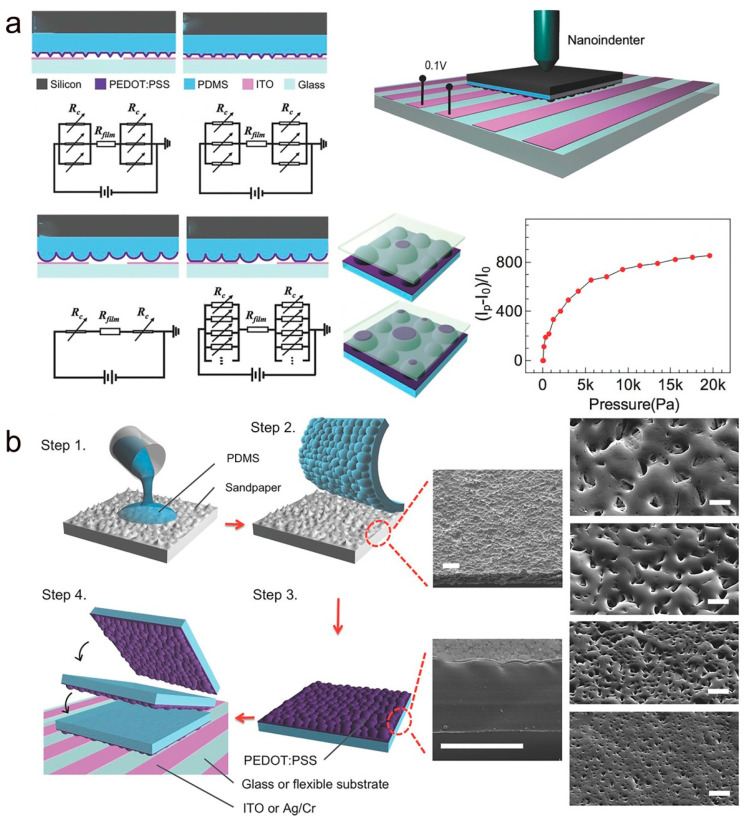
(**a**) Cross-sectional views of the piezoresistive sensors with different microstructures and in the corresponding equivalent circuit diagrams. (**b**) Flow chart of the preparation of the pressure sensor and SEM images of the microstructure of different specifications. (**a**,**b**) Reproduced with permission [90]. Copyright 2016, Wiley-VCH.

**Figure 4 molecules-28-01627-f004:**
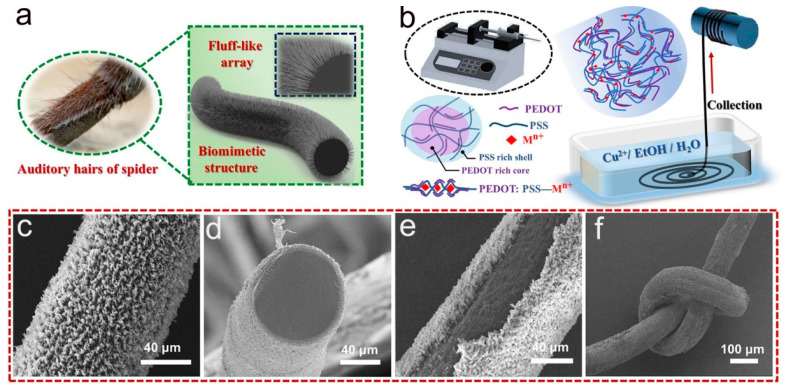
(**a**) PEDOT:PSS-Cu^2+^fiber. (**b**) PEDOT:PSS-Cu^2+^fiber’s manufacturing process. (**c**–**f**) SEM images of PEDOT:PSS-Cu^2+^fiber. (**a**–**f**) Reproduced with permission [91]. Copyright 2021, Elsevier.

**Figure 5 molecules-28-01627-f005:**
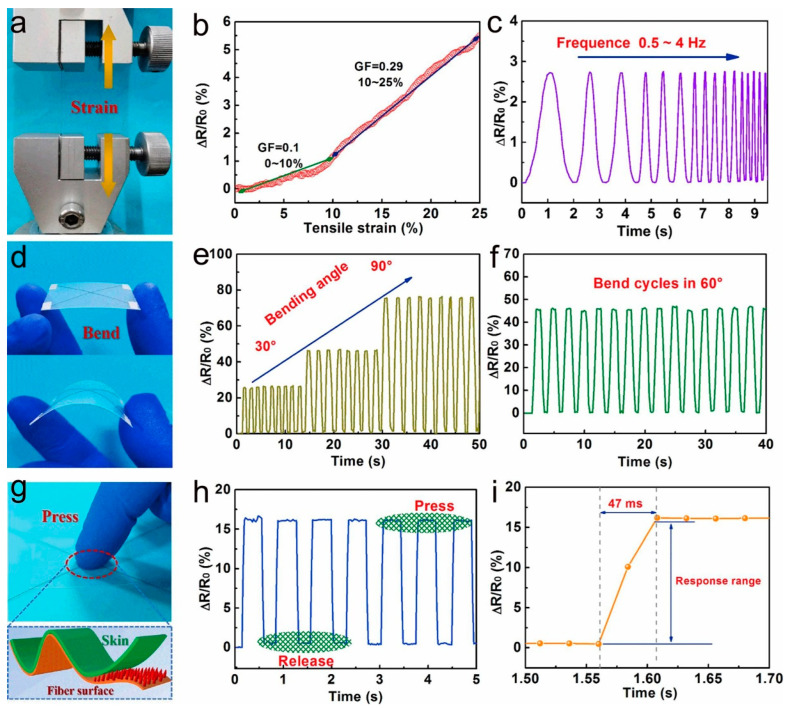
(**a**–**c**) The sensing performance of PEDOT:PSS-Cu^2+^ fiber-based tactile sensors during stretching. (**d**–**f**) The sensing performance of PEDOT:PSS-Cu^2+^ fiber-based tactile sensors during bending. (**g**–**i**) The sensing performance of PEDOT:PSS-Cu^2+^ fiber-based tactile sensors during compression. (**a**–**i**) Reproduced with permission [91]. Copyright 2021, Elsevier.

**Figure 6 molecules-28-01627-f006:**
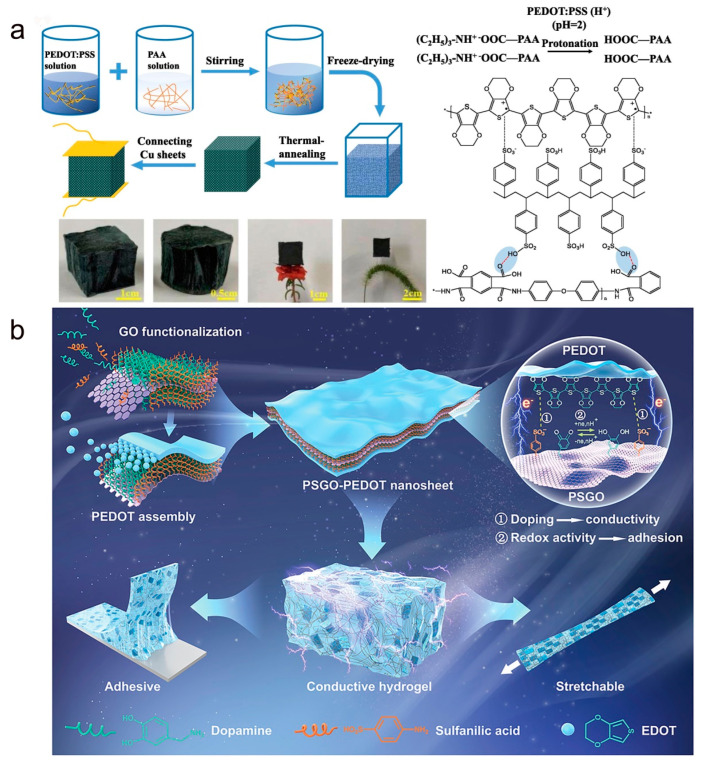
(**a**) Schematic diagram of PEDOT:PSS/PI aerogel manufacturing process. Reproduced with permission [79]. Copyright 2020, Elsevier. (**b**) Schematic diagram of PSGO-PEDOT nanosheets and their doping in hydrogels. Reproduced with permission [92]. Copyright 2020, Wiley-VCH.

**Figure 7 molecules-28-01627-f007:**
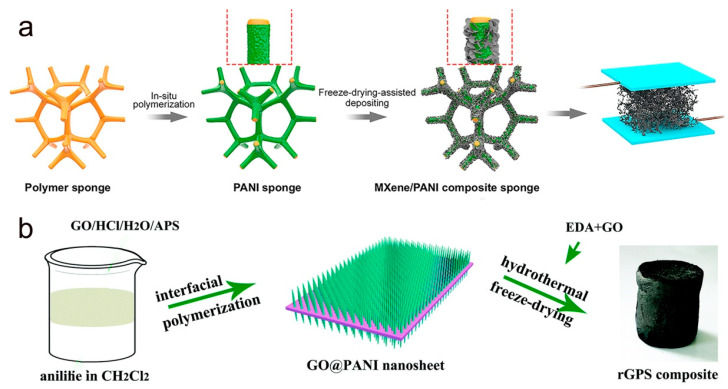
(**a**) Fabrication process of 3D porous MXene/PANI sponge. Reproduced with permission [81]. Copyright 2021, Elsevier. (**b**) Schematic diagram of the fabricated GO@PANI plate and rGPS composite. Reproduced with permission [89]. Copyright 2019, The Royal Society of Chemistry.

**Figure 8 molecules-28-01627-f008:**
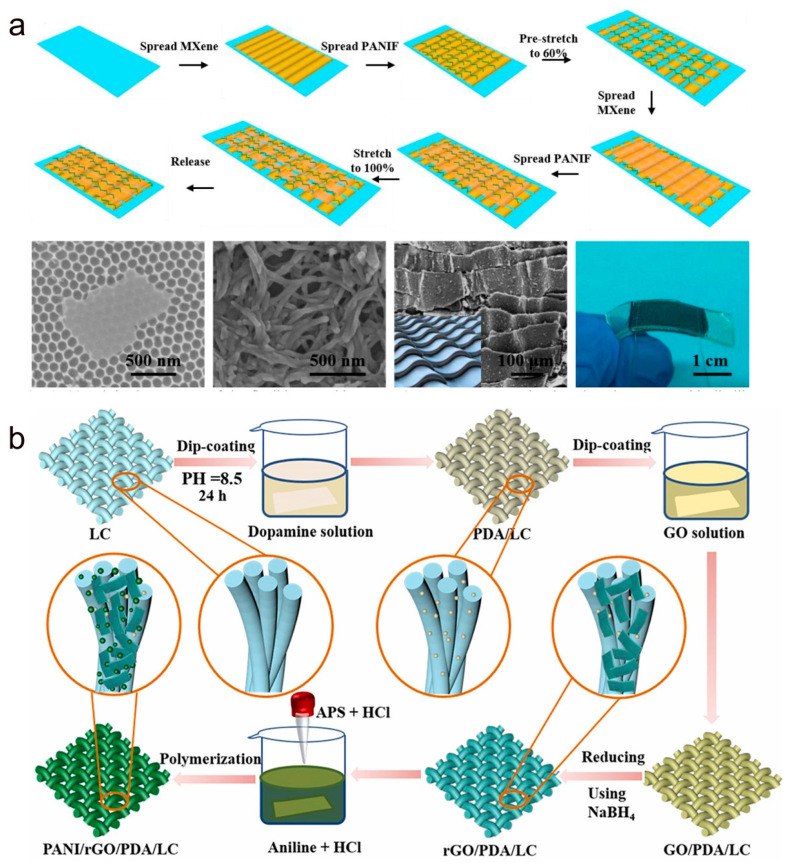
(**a**) Preparation process of the tile-like stacked layered microstructure of MXene/PANIF nanocomposite strain sensor. SEM images of the MXene sheet and PANIF sheet and photos of the sensor. Reproduced with permission [94]. Copyright 2020, Elsevier. (**b**) Schematic diagram of the manufacturing process of PANI/RGO/PDA/LC. Reproduced with permission [77]. Copyright 2020, Elsevier.

**Figure 9 molecules-28-01627-f009:**
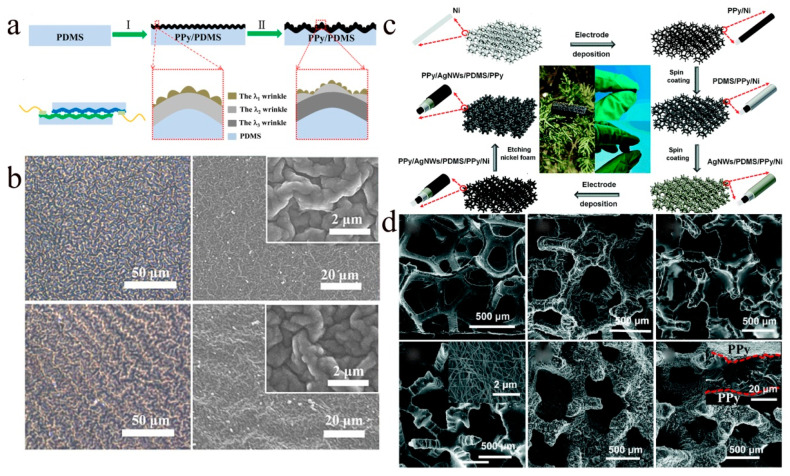
(**a**) Schematic diagram of PPy film with wrinkled microstructure. (**b**) Optical and SEM images of PPy films with nested wrinkled microstructures. The inset shows the magnified SEM image of the PPy film. (**a**,**b**) Reproduced with permission [100]. Copyright 2018, American Chemical Society. (**c**) Flow chart for the preparation of multi-scale microstructured porous PPy sponges. (**d**) SEM images of the morphological evolution during the preparation of PPy composites. (**c**,**d**) Reproduced with permission [101]. Copyright 2020, The Royal Society of Chemistry.

**Figure 10 molecules-28-01627-f010:**
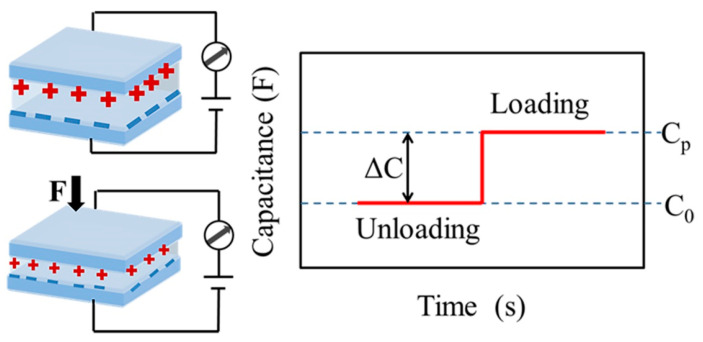
Work principle of capacitive pressure sensors.

**Figure 11 molecules-28-01627-f011:**
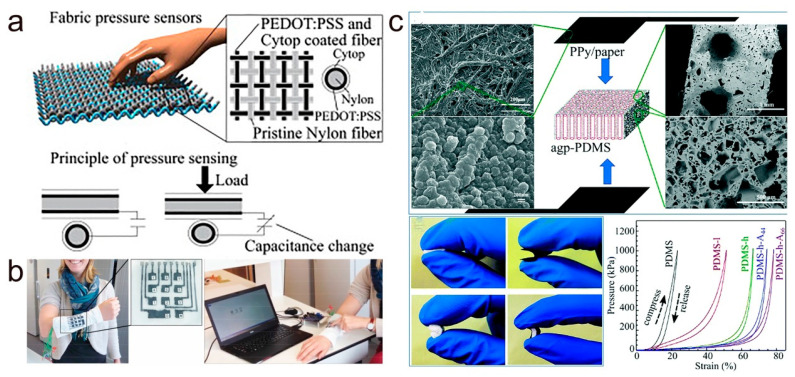
(**a**) Schematic diagram of the structure and sensing mechanism of the PEDOT:PSS-coated textile-based capacitive pressure sensor. Reproduced with permission [106]. Copyright 2015, Wiley-VCH. (**b**) Wearable keyboard based on PEDOT:PSS textile printing. Reproduced with permission [107]. Copyright 2015, Wiley-VCH. (**c**) Schematic diagram of the device structure of the PPy/paper/agp-PDMS-based capacitive pressure sensor and the pressure–strain relationship diagram. Reproduced with permission [108]. Copyright 2020, The Royal Society of Chemistry.

**Figure 12 molecules-28-01627-f012:**
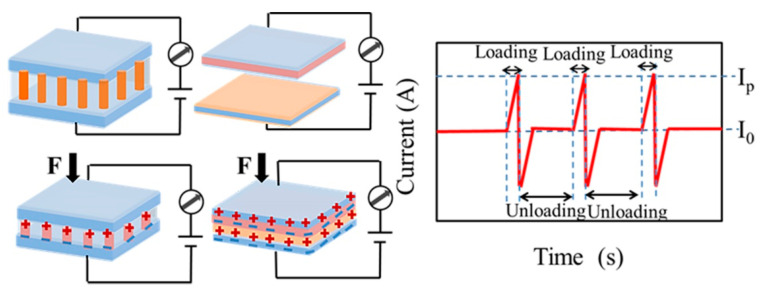
Work principle of piezoelectric and triboelectric pressure sensors.

**Figure 13 molecules-28-01627-f013:**
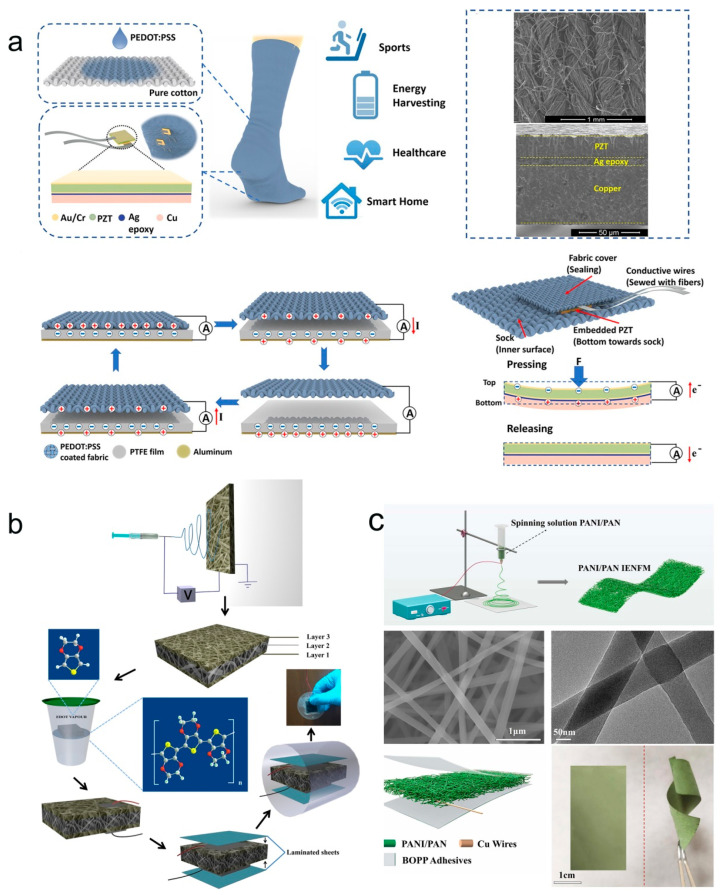
(**a**) Schematic and SEM images of the TENG and embedded PZT piezoelectric chips based on PEDOT:PSS-coated fabric. The schematic diagram of PEDOT:PSS TENG and PZT piezoelectric chip operation is shown at the bottom. Reproduced with permission [114]. Copyright 2019, American Chemical Society. (**b**) OPNG manufacturing process flow diagram. Reproduced with permission [119]. Copyright 2018, American Chemical Society. (**c**) Schematic and SEM images of electrostatic spinning of PANI/PAN nanofibers. The schematic diagram of the PANI/PAN NF piezoelectric nanogenerator and PANI/PAN NF optical photograph is shown at the bottom. Reproduced with permission [120]. Copyright 2022, Elsevier.

**Figure 14 molecules-28-01627-f014:**
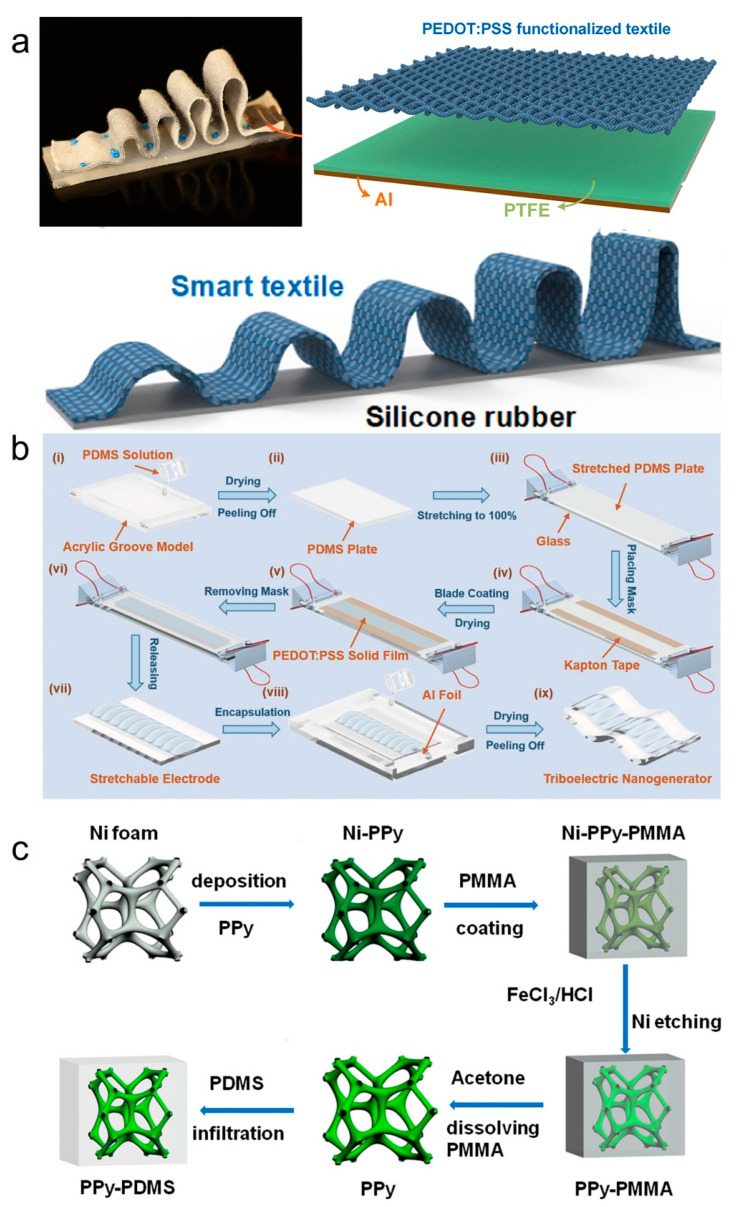
(**a**) Schematic diagram of the height-varying stretchable arch TENG strain sensor. Reproduced with permission [128]. Copyright 2018, Elsevier. (**b**) Schematic diagram of the transparent and stretchable PEDOT:PSS electrode-based TENG manufacturing process. Reproduced with permission [129]. Copyright 2018, Wiley-VCH. (**c**) Schematic diagram of the manufacturing process of PPy/PDMS foam. Reproduced with permission [133]. Copyright 2018, Elsevier.

**Figure 15 molecules-28-01627-f015:**
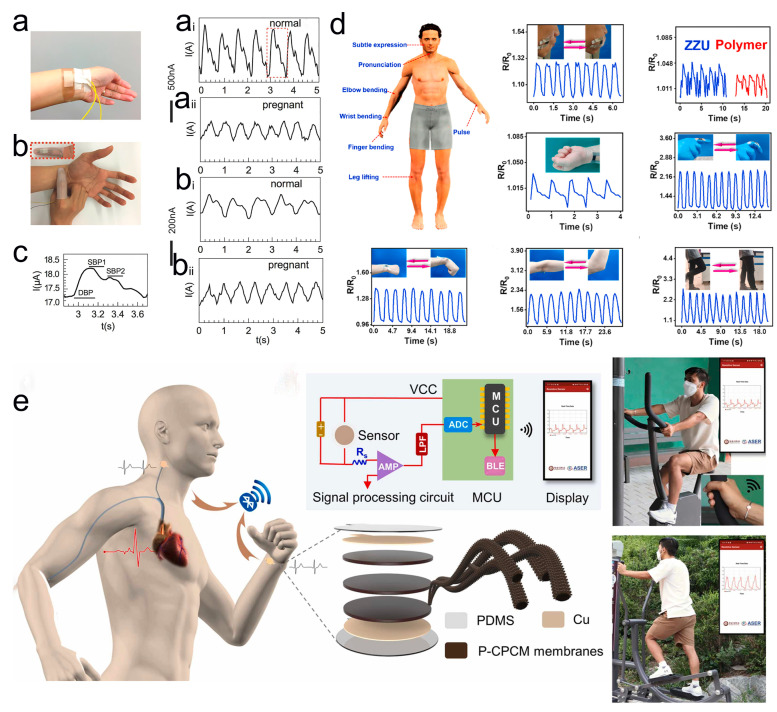
(**a**) Photograph of a pressure sensor fixed to a human wrist. Passive detection of output current on the wrist of normal (**a_i_**) and pregnant women (**a_ii_**) schematic. (**b**) Photo of the pressure sensor embedded in the fingertip glove. Schematic diagram of active detection of output current on the wrist of normal (**b_i_**) and pregnant women (**b_ii_**). (**c**) The enlarged view marked with a red wireframe in Figure (**a_i_**) has compression peaks and diastolic pressure lines typical of the program. (**a**–**c**) Reproduced with permission [90]. Copyright 2016, Wiley-VCH. (**d**) The sensors are fixed to different parts of the body to detect human movement schematically. Reproduced with permission [157]. Copyright 2022, Elsevier. (**e**) Schematic diagram of the sensor structure, wireless system mounted on the human body for real-time health monitoring. Reproduced with permission [158]. Copyright 2022, Elsevier.

**Figure 16 molecules-28-01627-f016:**
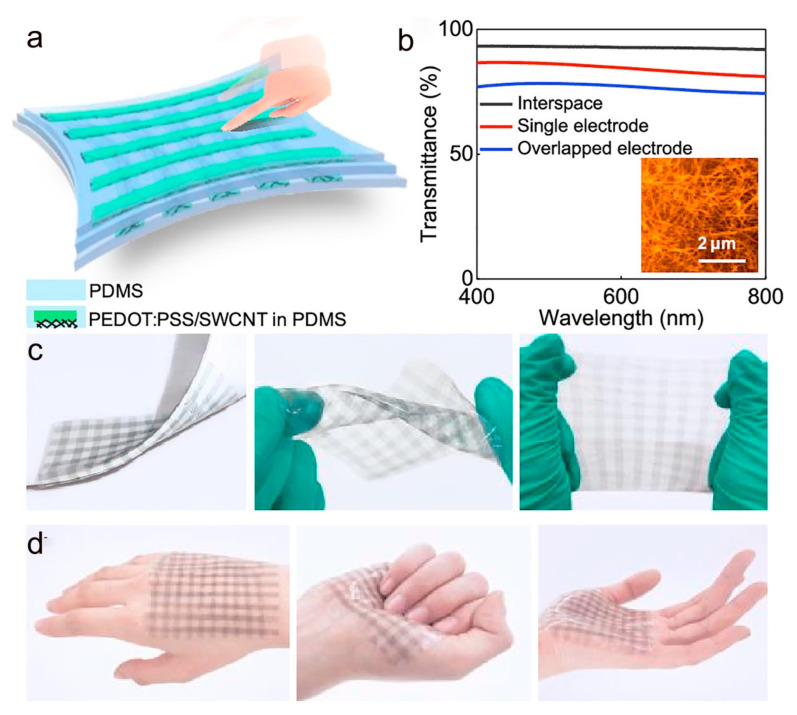
(**a**) Schematic diagram of the sensor structure based on PEDOT:PSS/SWCNT electrode. (**b**) Transmittance curve of the sensor; AFM images of PEDOT:PSS/SWCNT electrodes. (**c**) Photographs of electronic skin under folding, twisting, and stretching conditions. (**d**) An electronic skin photograph is attached to the palm. (**a**–**d**) Reproduced with permission [139]. Copyright 2020, American Chemical Society.

**Figure 17 molecules-28-01627-f017:**
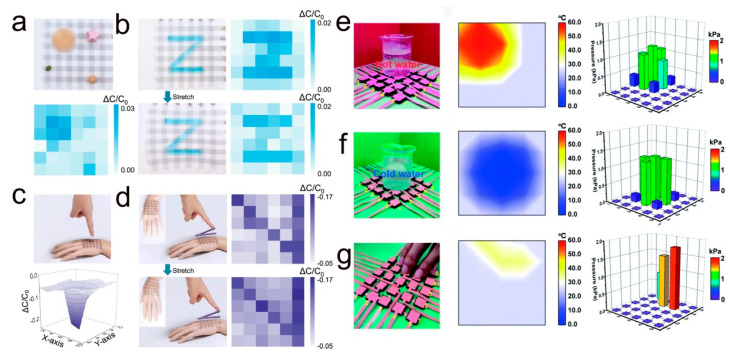
(**a**) Electronic skin photographs of placed coins, beans, and paper stars and the corresponding pressure distribution maps. (**b**) Photographs of the electronic skin and the corresponding pressure distribution of the Z-shaped object loaded in the initial state and at 20% strain, respectively. (**c**) Photographs of the finger as it approaches the electronic skin and the corresponding pressure space mapping. (**d**) Electronic skin monitoring of finger movement trajectory photos and corresponding pressure distribution. (**a**–**d**) Reproduced with permission [139]. Copyright 2020, American Chemical Society. (**e**–**g**) Photographs of a cup of hot water and a cup of cold water and a finger touching the touch e-skin and the corresponding temperature and pressure response graphs are placed, respectively. Reproduced with permission [142]. Copyright 2022, The Royal Society of Chemistry.

**Figure 18 molecules-28-01627-f018:**
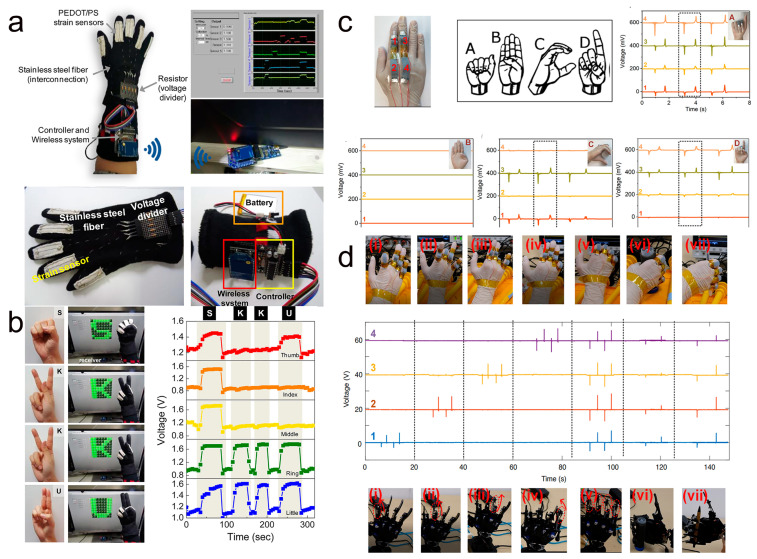
(**a**) Textile-based wearable human–machine interface smart glove and communication control system. (**b**) Recognition of gestures by wearable human–machine interface devices with corresponding output voltage signals. (**a**,**b**) Reproduced with permission [160]. Copyright 2017, American Chemical Society. (**c**,**d**) Self-powered sensor based on PEDOT:PSS coating for sign language recognition and robot hand control. Reproduced with permission [128]. Copyright 2018, Elsevier.

**Figure 19 molecules-28-01627-f019:**
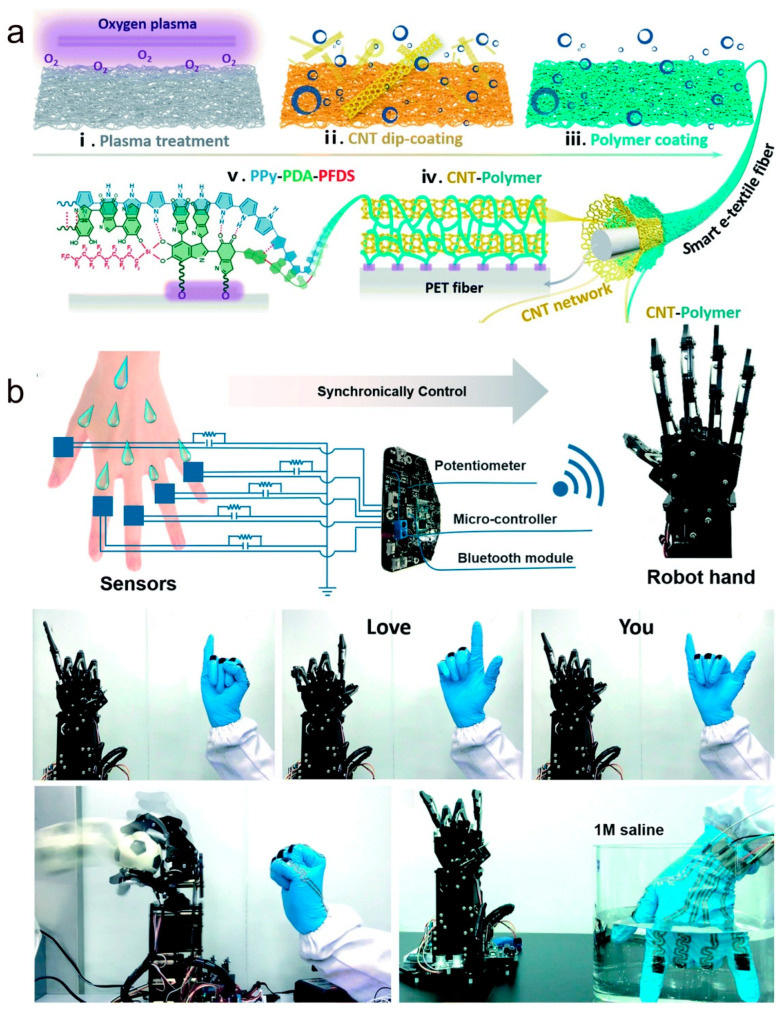
(**a**) Schematic diagram of the preparation and structure of smart textile fibers with a “steel-concrete” layered structure. (**b**) Synchronized control of a robotic hand by human hand movements based on wearable smart textile sensors. (**a**,**b**) Reproduced with permission [144]. Copyright 2019, The Royal Society of Chemistry.

**Figure 20 molecules-28-01627-f020:**
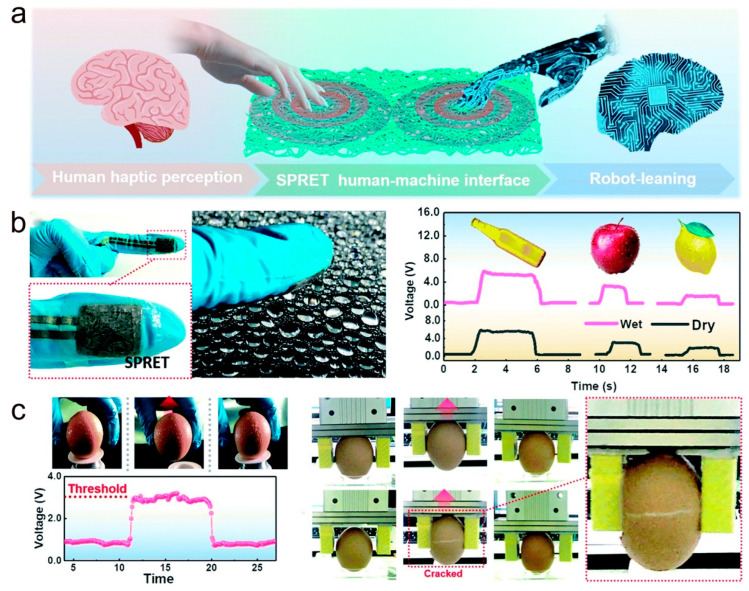
(**a**) Schematic diagram of the robot training system. (**b**) Photographs of sensor fingers based on a “steel-concrete” layered structure, grasping the electrical signals in response to bottles, apples, and lemons, respectively. (**c**) The robot learns to grasp eggs by reading human-sensing information. (**a**–**c**) Reproduced with permission [144]. Copyright 2019, The Royal Society of Chemistry.

**Figure 21 molecules-28-01627-f021:**
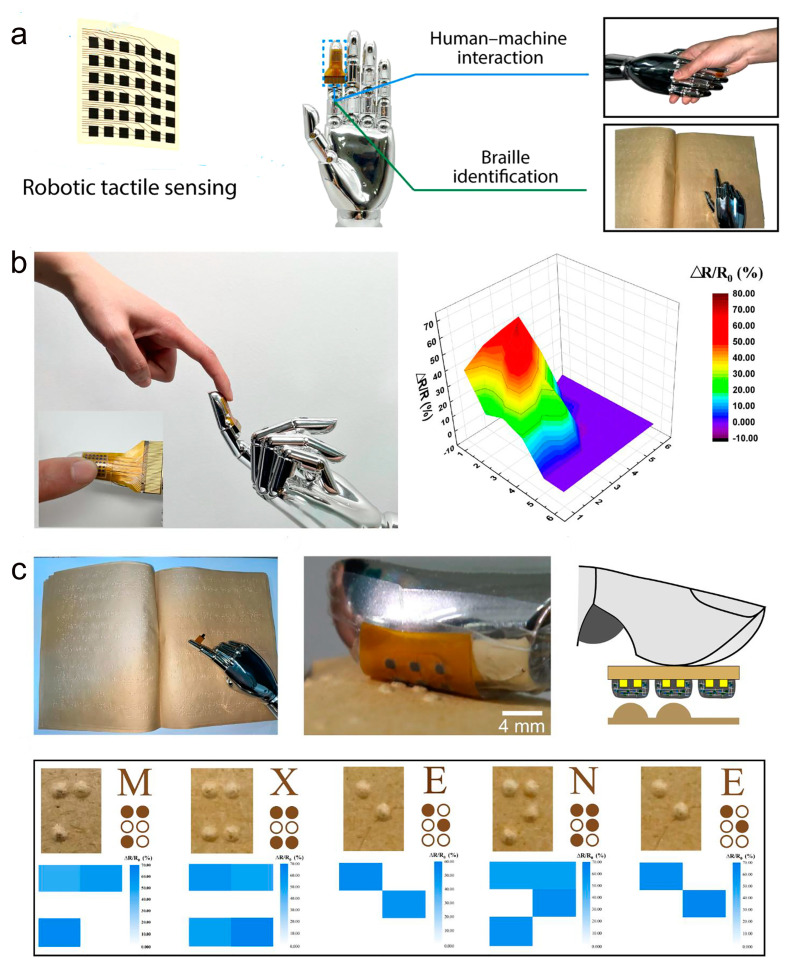
(**a**) MPCA haptic sensor microarrays for human–machine interaction and robotic tactile sensing. (**b**) The artificial tactile interface at the fingertips of the robot senses tactile stimuli from human fingers and displays spatial pressure mapping maps. (**c**) Manual tactile interface for Braille recognition. (**a**–**c**) Reproduced with permission [140]. Copyright 2022, American Chemical Society.

**Table 1 molecules-28-01627-t001:** A summary of the device performances of pressure sensors.

Materials	Sensing Mechanism	Minimum Detection	Response Time	Sensitivity	Ref.
PEDOT:PSS/PDMS	Piezoresistive	7.14 Pa	200 μs	642.5 kPa^−1^	[30]
PEDOT:PSS/PI	Piezoresistive	NA	NA	0.054 kPa^−1^	[79]
PEDOT: PSS-Cu^2+^ fibers	Piezoresistive	82 Pa	47 ms	0.34 kPa^−1^	[91]
PEDOT:PSS@Melamine Conductive Sponge	Piezoresistive	NA	NA	GF = 0.41	[87]
PEDOT:PSS/PAAMPSA	Piezoresistive	NA	19 ms	164.5 kPa^−1^	[134]
PEDOT:PSS/CNT/Ecoflex	Piezoresistive	NA	NA	4.11 kPa^−1^	[135]
PEDOT:PSS/CNT/PDMS	Piezoresistive	NA	2.5 ms	2.32 kPa^−1^	[135]
PEDOT:PSS/PDMS	Piezoresistive	NA	90 ms	21 kPa^−1^	[136]
PEDOT:PSS/PUD micropyramid	Piezoresistive	23 Pa	NA	10.3 kPa^−1^	[2]
PEDOT:PSS/PPy	Piezoresistive	NA	0.36 ms	0.58 kPa^−1^	[137]
Ti_3_C_2_T_x_-PEDOT:PSS/PDMS	Piezoresistive	NA	200 ms	133.32 kPa^−1^	[138]
PEDOT:PSS/SWCNT/PDMS	Piezoresistive	0.6 Pa	82 ms	0.1 kPa^−1^	[139]
MXene/PEDOT:PSS Composite Aerogel	Piezoresistive	NA	106 ms	26.65 kPa^−1^	[140]
PEDOT:PSS-Modified Polyurethane Foam	Piezoresistive	NA	NA	0.3 kPa^−1^	[141]
PEDOT:PSS/CNT@PDA@PDMS	Piezoresistive	NA	170 ms	1.97% kPa^−1^	[142]
PEDOT:PSS/PDMS	Piezoresistive	NA	0.15 ms	851 kPa^−1^	[90]
PPy/rGO@carbonized PU	Piezoresistive	NA	69 ms	0.635 kPa^−1^	[143]
PPy/PDMS	Piezoresistive	1 Pa	20 ms	19.32 kPa^−1^	[100]
rGO-PANI sponge	Piezoresistive	NA	50 ms	0.77 kPa^−1^	[89]
elastic microstructured conducting polymer	Piezoresistive	<1 Pa	47 ms	133.1 kPa^−1^	[88]
MXene/PANIF	Piezoresistive	0.1538%	NA	2369.1	[94]
PEDOT:PSS/Cytop	Capacitive	0.01 N	NA	0.369 N^−1^	[106]
PPy/filter paper	Capacitive	5 Pa	NA	1.15 kPa^−1^	[144]
wrinkled PEDOT:PSS/PDMS	Triboelectric	2 kPa	200 ms	0.08 kPa^−1^	[129]
PPy-PDMS	Triboelectric	0.6 kPa	NA	12.61 pF·kPa^−1^	[133]
PEDOT:PSS-coated fabric	Triboelectric	10 kPa	NA	0.228 V/N	[114]
PVDF/PVDF/PEDOT	Piezoelectric	1 kPa	NA	6.5 kPa^−1^	[119]
PANI/PAN	Piezoelectric	0.1 N	66 ms	1.71 VN^−1^	[120]

PEDOT: PSS: poly(3,4-ethylenedioxythiophene):poly(styrenesulfonate); PDMS: polydimethylsiloxane; PI: polyimide; PAAMPSA: poly(2-acrylamido-2-methyl-1-propanesulfonic acid; CNT: carbon nanotubes; PUD: polyurethane dispersion; PPy: polypyrrole; SWCNT: single-walled carbon nanotube; PDA: poly(dopamine); PU: polyurethane; rGO: reduced graphene Oxide; PANI: polyaniline; PANIF: polyaniline fiber; PAN: polyacrylonitrile; PVDF: Polyvinylidene fluoride.

**Table 2 molecules-28-01627-t002:** The device performance of other polymer composite-based pressure sensors.

Materials	Sensing Mechanism	Minimum Detection	Response Time	Sensitivity	Ref.
Ti_3_C_2_T_x_/PDMS	Piezoresistive	4.4 Pa	<130 ms	151.4 kPa^−1^	[145]
MXene/SWCNT/PVP	Piezoresistive	0.69 Pa	48 ms	165.35 kPa^−1^	[146]
PVA/SA/MXene	Piezoresistive	0.2%	NA	GF = 0.97	[147]
rGO/PVDF	Piezoresistive	1.3 Pa	20 ms	47.7 kPa^−1^	[148]
rGO/PDMS	Piezoresistive	16 Pa	80 ms	25.1 kPa^−1^	[149]
SSNPs/PU	Piezoresistive	300 Pa	30 ms	2.46 kPa^−1^	[150]
AgNWs/PDMS	Piezoresistive	NA	20 ms	0.03 kPa^−1^	[151]
CB/TPU	Piezoresistive	10 Pa	18 ms	5.54 kPa^−1^	[152]
Graphite/PDMS	Piezoresistive	0.9 Pa	8 ms	64.3 kPa^−1^	[41]
AuNWs/PDMS	Piezoresistive	NA	10 ms	23 kPa^−1^	[45]
CNT/PDMS	Piezoresistive	0.2 Pa	40 ms	15.1 kPa^−1^	[68]
MXene/PVDF-TrFE	Capacitive	1.5 Pa	150 ms	0.51 kPa^−1^	[61]
Porous PDMS/air gap	Capacitive	2.5 Pa	NA	0.7 kPa^−1^	[153]
LM-NP/PDMS composite	Triboelectric	NA	NA	2.52 V·kPa^−1^	[154]
MXene-fiber	Piezoelectric	0.1 N	5 ms	51.5 mV/N	[155]
P(VDF-TrFE) nanofiber	Piezoelectric	0.1 Pa	NA	0.41 V Pa^−1^	[156]

PDMS: polydimethylsiloxane; SWCNT: single-walled carbon nanotube; PVP: polyvinylpyrrolidone; SA: sodium alginate; rGO: reduced Graphene Oxide; PVDF: Polyvinylidene fluoride; SSNPs: sea-urchin shaped metal nanoparticles; PU: polyurethane; AgNWs: silver nanowires; CB: carbon blacks; TPU: thermoplastic urethane elastomer; AuNWs: gold nanowires; CNT: carbon nanotubes; PVDF-TrFE: poly (vinylene fluoride-trifluoroethylene); LM-NP: Liquid-Metal-Nanoparticle.

## Data Availability

The data is available in section “MDPI Research Data Policies” at https://www.mdpi.com/ethics.

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
