# Peer review of "Conjugated Polymer-Based Nanocomposites for Pressure Sensors"

_molecules, 2023, doi:10.3390/molecules28041627_

Round 1

Reviewer 1 Report

I think this is a very nice review manuscript on conjugated polymer nanocomposites-based pressure sensors. The authors summarized the types of pressure sensors, discussed the sensing mechanism and the structure very clearly, and the figures are attractive. However, the manuscript has some issues, which need to be fully addressed, before it can be considered for publication.

1. The authors need to add more discussion of wash ability of the (textile) sensor.

2. The authors need to explain the advantages of conjugated polymers over other materials in the literature. A table was recommended to be added in the manuscript.

Author Response

Responses to the reviewers’ comments:

We would like to thank you for your time and insightful comments on our manuscript. We have revised the manuscript according to your comments/suggestions.

Reviewer #1:

Comment 1: The authors need to add more discussion of wash ability of the (textile) sensor.

Authors’ Response: Thank you very much for your comments. Following your suggestion, we have added some relevant discussion on the washing capacity of textile sensors to lines 278, 654 of the manuscript. For example, in the discussion of reference 143: thanks to a polymer coating on the textile surface and a special structure, the textile sensors was machine washed for 3 cycles and subjected to several hours of agitated washing with negligible changes in sensitivity, and can also accurately detect a range of human movement and physiological signals. The related part has been highlighted in the revised manuscript.

Comment 2: The authors need to explain the advantages of conjugated polymers over other materials in the literature. A table was recommended to be added in the manuscript.

Authors’ Response: Thank you very much for your comments and insightful suggestions. We compare the advantages of conjugated polymers over other materials in the literature. According to your suggestion, the device performances of pressure sensors based on other polymer composites beyond conjugated polymers are listed in Table 2. We revised the related part in the manuscript as “Furthermore, to show the advance of conjugated polymers based pressure sensors, we compare their device performance with those of the state-of-art pressure sensors based on other polymer composites as shown in Table 2.” The related part has been highlighted in the revised manuscript.

Table 2. The device performance of other polymer composite based pressure sensors.

Materials

Sensing mechanism

Minimum detection

Response time

Sensitivity

Ref.

Ti3C2Tx/PDMS

Piezoresistive

4.4 Pa

<130 ms

151.4 kPa-1

[150]

MXene/SWCNT/PVP

Piezoresistive

0.69 Pa

48 ms

165.35 kPa−1

[151]

PVA/SA/MXene

Piezoresistive

0.2%

NA

GF = 0.97

[152]

rGO/PVDF

Piezoresistive

1.3 Pa

20 ms

47.7 kPa−1

[153]

rGO/PDMS

Piezoresistive

16 Pa

80 ms

25.1 kPa-1

[154]

SSNPs/PU

Piezoresistive

300 Pa

30 ms

2.46 kPa−1

[155]

AgNWs/PDMS

Piezoresistive

NA

20 ms

0.03 kPa−1

[156]

CB/TPU

Piezoresistive

10 Pa

18 ms

5.54 kPa−1

[157]

Graphite/PDMS

Piezoresistive

0.9 Pa

8 ms

64.3 kPa−1

[158]

AuNWs/PDMS

Piezoresistive

NA

10 ms

23 kPa−1

[159]

CNT/PDMS

Piezoresistive

0.2 Pa

40 ms

15.1 kPa−1

[160]

MXene/PVDF-TrFE

Capacitive

1.5 Pa

150 ms

0.51 kPa−1

[161]

Porous PDMS/air gap

Capacitive

2.5 Pa

NA

0.7 kPa−1

[162]

LM-NP/PDMS composite

Triboelectric

NA

NA

2.52 V·kPa−1

[163]

MXene-fiber

Piezoelectric

0.1 N

5 ms

51.5 mV/N

[164]

P(VDF-TrFE) nanofiber

Piezoelectric

0.1 Pa

NA

0.41 V Pa−1

[165]

PDMS: polydimethylsiloxane; SWCNT: single-walled carbon nanotube; PVP: polyvinylpyrrolidone; SA: sodium alginate; rGO: reduced Graphene Oxide; PVDF: Polyvinylidene fluoride; SSNPs: sea-urchin shaped metal nanoparticles; PU: polyurethane; AgNWs: silver nanowires; CB: carbon blacks; TPU: thermoplastic urethane elastomer; AuNWs: gold nanowires; CNT: carbon nanotubes; PVDF-TrFE: poly (vinylene fluoride-trifluoroethylene); LM-NP: Liquid-Metal-Nanoparticle.

Reviewer 2 Report

Flexible pressure sensors play a more and more important role in human daily life. In this review, the author systematically and progressively elaborated the parameters and mechanism of typical pressure sensors and then introduced four typical conjugated polymer-based pressure sensors. After that, the author compared the main performance parameter of all the recently reported conjugated polymer pressure sensor devices. At last, the author introduced the wide-range application of conjugated polymers based pressure sensors.

This review was finished with very high quality, which means it can almost be published in the current version. However, there are still some detailed errors in this review, which need to be carefully corrected before publication. If the following problems are well-addressed, this reviewer believes that the essential contribution of this review is important for the research of conjugated polymer pressure sensors.

1, There are at least two grammatical and spelling errors in the manuscript, such as, in line14, “is” should be removed before“ the tactile…”; in line 70, “tothe” should be “ to the”; in line 97 “helpfulfor” should be “ helpful for”.

2, In line 230, Use “hydrogen and Pi-Pi bonds” is not very accurate, which should be “hydrogen bond and Pi-Pi stacking interaction”.

3, A space is required between the number and the unit, such as, in lines 296, 297, and also in Table 1.

4, Authors need to unify the format of all references, such as whether to use both the starting and ending page numbers (Ref. 55, 135, 139), and whether to abbreviate the journal name (Ref. 32, 44, 45). The volume of Ref. 54 shows “n/a“, which is an obvious error.

Author Response

Responses to the reviewers’ comments:

We would like to thank you for your time and insightful comments on our manuscript. We have revised the manuscript according to your comments/suggestions.

Reviewer #2: 

Comment 1: There are at least two grammatical and spelling errors in the manuscript, such as, in line14, “is” should be removed before “the tactile…”; in line 70, “tothe” should be “ to the”; in line 97 “helpfulfor” should be “ helpful for”.

Authors’ Response: Thank you very much for your comments. Based on your comments, we have corrected the grammatical and spelling errors that you pointed out in the manuscript. We also checked and revised the typos thorough the manuscript. The related revisions have been highlighted in the revised manuscript.

Comment 2: In line 230, Use “hydrogen and Pi-Pi bonds” is not very accurate, which should be “hydrogen bond and Pi-Pi stacking interaction”.

Authors’ Response: Thank you very much for your comments and insightful suggestions. Based on your suggestion, we have revised the expression “hydrogen and Pi-Pi bonds….” in the manuscript as “hydrogen bond and π-π stacking interaction… ”. The related part has been highlighted in the revised manuscript.

Comment 3: A space is required between the number and the unit, such as, in lines 296, 297, and also in Table 1.

Authors’ Response: Thank you very much for your suggestions. Following your suggestion, we double-checked the spaces between numbers and units in the manuscript and made revisions. The related parts have been highlighted in the revised manuscript.

Comment 4: Authors need to unify the format of all references, such as whether to use both the starting and ending page numbers (Ref. 55, 135, 139), and whether to abbreviate the journal name (Ref. 32, 44, 45). The volume of Ref. 54 shows “n/a“, which is an obvious error.

Authors’ Response: Thank you very much for your suggestions. Based on your suggestions, we have standardized the format of references in the manuscript according to the journal requirements and corrected some errors. The related revisions have been highlighted in the revised manuscript.